# The Use of IoT Technology in Smart Cities and Smart Villages: Similarities, Differences, and Future Prospects

**DOI:** 10.3390/s20143897

**Published:** 2020-07-13

**Authors:** Nina Cvar, Jure Trilar, Andrej Kos, Mojca Volk, Emilija Stojmenova Duh

**Affiliations:** Faculty of Electrical Engineering, University of Ljubljana, Tržaška Cesta 25, 1000 Ljubljana, Slovenia; nina.cvar@fe.uni-lj.si (N.C.); jure.trilar@fe.uni-lj.si (J.T.); andrej.kos@fe.uni-lj.si (A.K.); mojca.volk@fe.uni-lj.si (M.V.)

**Keywords:** IoT, sensors, technology, smart cities, smart villages, digital innovation ecosystems, public value, co-production, place-based approach, context-based approach

## Abstract

Initially, the concept of Smart Cities (urban settlement) originated from the Internet of Things (IoT) technology, however, the use of IoT technology can be extended to the concept of Smart Villages (rural settlement) as well, improving the life of the villagers, and the communities as a whole. Yet, the rural settlements have slightly different requirements than the urban like settlements. If application of IoT in Smart Cities can be characterized by densification of IoT to day-to-day life, following cities’ structural characteristics of being densely settled places, IoT empowered Smart Villages are usually a system of dispersion and deficiency. In this manner, this research paper will address and discuss different application areas of IoT technology, identifying differences, but also similarities in both ecosystems, while trying to illuminate the standardization efforts that can be applicable in both contexts. In our text we will propose the following IoT application domains, which will also serve as a base for research on smart villages: 1. Natural Resources and Energy, 2. Transport and Mobility, 3. Smart Building, 4. Daily Life, 5. Government, and 6. Economy and Society. By providing an overview of technical solutions that support smart solutions in Smart Cities and Smart Villages this research paper will evaluate how, with IoT empowered Smart Villages and Smart Cities, an overall improvement of quality of life of their inhabitants can be achieved.

## 1. Introduction

The Internet of Things (IoT) technology can be regarded as the building block for next generation Smart Cities (SC) due to its potential in exploiting sustainable information and communication technologies [1]. Initially the concept of SCs (urban settlements) has originated from the Internet of Things (IoT) technology. However, the use of IoT (technology) can be extended to the Smart Villages (SV) in rural settlement as well, improving the life of the communities as a whole, demonstrating that the IoT (technology) is not limited to either context or does not exclude the understanding that there exists a difference in priorities of planning these solutions on distinct levels and contexts [2].

By providing a transdisciplinary analysis, this work will examine the role of Internet of Things (IoT) technology in SCs and SVs. To achieve this objective, this research will address and discuss different application areas of the IoT (technology), identifying differences as well as similarities in both digital innovation ecosystems, while trying to illuminate the standardization efforts that can be applicable in both contexts. 

There is an immense quantity of academic resources available on IoT (technology) in SCs, whereas on the other hand the resources on IoT (technology) implementations on SVs are relatively sparse, many of them are not even well detailed. To reconsider the similarities, differences, and future prospects of the use of the IoT (technology) in SCs and SVs, this study has drawn from disparate resources. 

In terms of the broad spectrum of solutions in SCs, we have followed the state-of-the-art review works by Pablo Chamoso et al. [3] and Alavi et al. [1]. On the other hand, studying the use of the IoT (technology) in SVs proved to be a lot more challenging. In this manner, a theoretical framework on SVs was needed and only then the research on the IoT (technology) use in rural areas followed. 

Regarding the analysis of the social aspects of SVs, the works of Zavratnik et al. [4], Provenzano et al. [5] and Hosseini et al. [6], Ruiz-Mallén [7], and the policies of the EU and the work of The international association Rurality–Environment–Development (R.E.D.) rendered the necessary analytical grounds.

The study of the social dimension of the application of the IoT (technology) in SCs and SVs endowed this research with the transdisciplinary character, concretely by the concept of the digital innovation ecosystem and by the work of Visvizi and Lytras [8] which via design of the nested-clusters model demonstrates that the smart cities research can contribute to research on smart villages as well. 

Overall, the literature on the IoT (technology) solutions in SVs presented us with an important insight on the differences between Europe’s rural areas and undeveloped regions, attesting the role of the socio-economic, cultural, and policy characteristics of the specific regions. 

Along these lines, this study has grounded its research on Kaur’s argument [2] on the IoT (technology) not being limited to the context of use, although not neglecting different requirements in relation to the application of IoT (technology) in both studied contexts. 

Obtained data from the study of the existing literature and data gathered from our research work within an array of conducted projects (domestic and international) led us to work on three constitutive parts of the IoT (technology) structure: Deriving from the work of Al-Fuqaha et al. [9], we have identified the necessary components of the IoT systems architecture, listing identification, sensing, communication, computation, services, and semantics (illustrated in Figure 1);Following International Organization for Standardization (ISO), International Telecommunication Union (ITU), European Telecommunications Standards Institute (ETSI), United Nations (UN) and Institute of Electrical and Electronics Engineers (IEEE, a list of standards and indicators directed at the digital innovation ecosystem of SCs have been singled out, proposing that they could be used within the digital innovation ecosystem of the Smart Village context as well;Similarities rather than disparities between both digital innovation ecosystems have been exemplified by the design of list of the IoT application domains with distinct service fields and use-cases, demonstrating that the use cases described in literature on SCs could be found within the digital innovation ecosystem of SVs too. The indicated list has been based on the inspection of numerous works/studies, with an aim to produce a conceptual roadmap that could provide us with the guidelines for conducting future research.

To encapsulate, the general contributions/objectives of this research are the following:Evaluating the role of the Internet of Things (IoT) technology in both digital innovation ecosystems;Providing a transdisciplinary study of the research topic in question due to its complexity, which demands sharing and creating knowledge from a variety of disciplines;Introducing a conceptual framework that will assure further research guidance and facilitate production of knowledge by interaction of different disciplines.

To achieve the objective of identifying similarities and differences of both digital innovation ecosystems, this work has been structured as follows: the first section puts down a description of the methodology used in this research. The following two sections address both digital innovation ecosystems and indicate basic definitions of the Smart City and the Smart Village. The overview of the IoT (technology) in both digital innovation ecosystems is discussed in Section 5. In this section also an IoT (technology) structure is proposed. Findings of this research are discussed in the discussion sector and final remarks can be found in the concluding sector. 

## 2. Methodology

To query the IoT (technology) in SCs and SVs, this study is based on three methodological phases: 1. An analysis of the literature to identify and subsequently construct key theoretical framework of the studies field and systematization of data obtained from the conducted domestic and international projects; 2. Develop and design appropriate conceptual framework and provide a suitable visualization of the conducted research; 3. Synthesis and the concluding analysis. In this manner, we have formulated two research questions: Both digital innovation ecosystems have their own structural socio-economic features as well as geographical distinctions, which need to be taken into account when designing IoT solutions in both ecosystems.Same IoT technology can be used in both digital innovation ecosystems: digital innovation ecosystem of SCs and digital innovation ecosystem of SVs. In an IoT technology domain the use of IoT architecture components are application-specific alongside the acceptance that contributing factor for connectivity is based on a difference in topographical features between the two ecosystems.

## 3. Digital Innovation Ecosystem of Smart Cities 

### 3.1. Definition of the Concept of Digital Innovation Ecosystem 

We will propose that for both SCs and SVs and their relation to IoT (technology), the concept of a digital innovation ecosystem is applied: first, to underscore the complexity of the digital transformation with all its accompanying phenomena, including the application of IoT technology, and secondly, to demonstrate the structural uniqueness of both ecosystems. By addressing dispersion and deficiency from the perspective of a digital innovation ecosystem, structural heterogeneity of both systems is underlined and generalization with potential oversimplification is avoided, as both systems are structurally unique and heterogeneous even from within. In this vein, this research paper will address and discuss different application areas of IoT (technology), identifying differences, but also similarities in both digital innovation ecosystems. Via this particular concept, both SC and SV will emerge as a complex web of interdependent entities and relationships between them, underlining their lively dynamics not just within but also to the outside. Referring to Wang, a multilevel model of the ecosystem with its layers, demonstrates itself to be particularly efficient, as it enables formulation of a comprehensive theory of digital innovations, with the ability of not losing the complexity of each proposed entity. 

Exactly where does the term of an ecosystem come from? Following Wang via Siegele, around twenty-five years ago, James Moore popularized the term of an ecosystem in business and management [10] (p. 2). Since then, this term and other ecological metaphors have been frequently appearing in discussions about digital innovations, referring to the broader environment, with various interacting stakeholders. According to Wang [10], the importance of ecological perspective is its ability of going beyond the dominant focus of IT innovation studies on the innovator or on the innovation, therefore being able to conceptualize multiple stakeholders and multiple innovations almost in a holistic set, just like a biological ecosystem consisting of different species and natural resources. By applying this particular perspective, it is possible to conceptually link “diverse entities, processes, products, services, organizations, industries, communities, as they draw on resources, including technology, attention, and knowledge, to create and realize the value of digital innovation” [10] (p. 1).

We argue that this conceptualization corresponds to authors’ current fieldwork within an Interreg project Catalysing Regions in Peripheral and Emerging Europe towards Digital Innovation Ecosystems (CARPE DIGEM) which actively works on creating functioning Digital Innovation Ecosystems (DIGEM). Within this project, the concept of the digital innovation ecosystem is understood as a complex system of various actors having different roles, interacting in mutual interdependence, constantly learning how to interact effectively. 

The conceptual proximity between both conceptualizations can also be found in the way, how the indicated Interreg project comprehends the structure of its digital innovation ecosystem: it is characterized by the multilevel framework of structures, strategies, tools and people [11] “that both complement and enhance the impact of the technology itself; in competences and skills, in organizational change, in new processes, business and governance models, and also in the intellectual assets that help create value from the new technologies.”.

To sum up, we have introduced the concept of a digital innovation ecosystem based on the sources: the first comes from the ecological perspective and the second originates from our field action work within an Interreg project of CARPE DIGEM. 

The presented concepts functions as the conceptual cornerstone for contextualizing the differences and similarities between SCs and SVs, their relation to the IoT technology and a multilevel conceptual framework of both terms is emphasized, allowing this text to underline to study of SCs and SVs’ digital innovation ecosystems. Let us now introduce in greater detail the SC’s digital innovation ecosystem. 

### 3.2. Contextualization, Taxonomy, and Smart Cities as Public Value

With more and more people living in cities—according to UN, around 2.5 billion more people will be living in cities by 2050 [12]—new personal and communal needs have come to the fore, running from complexed social, economic, and management dimensions. From the 1990s onwards, some of those challenges have started being addressed through the lenses of the concept of a SC, when, according to Hosseini et al. [6] the label of a SC first occurred, carrying strong technical connotations due to application of the new information and communication technologies (ICT) to cities, emphasizing their technological developments and digitalization. 

Yet, as suggested by the Hosseini et al. [6] O’Grady and O’Hare [13], Batty et al. [14] Albino et al. [15] etc., over the years, ICT have been evoked in such a way to improve urban systems and thus quality of life. Since then, the SC has become an example of an intertwined framework of ICT in such a way that the cities are able to develop and deploy different practices by which growing urbanization challenges can be efficiently addressed. 

However, following O’Grady and O’Hare, [3] a SC signifies ways of use, which are not always consistent; in fact, the above indicated authors are claiming that a one-size-fits-all definition of a smart city does not exist.

To minimize this conceptual fuzziness, in this text we are proposing a taxonomy of the SCs which divides existing definitions into three large groups; in this vein, the first group is characterized by the focus on relation between the SC and the broader environment, putting forward a conceptual perspective, which we will define as an ecosystem perspective. This perspective is best represented by Giffinger et al. defining SC as “a city well performing in a forward-looking way in economy, people, governance, mobility, environment, and living, built on the smart combination of endowments and activities of self-decisive, independent and aware citizens”[6] (p. 11). 

If the first group emphasizes the conjunction of the smart city with its environment, the second group of definitions puts more focus on the role of ICT in SCs, as, to quote Cohen and Muñoz [7] “smart cities use information and communication technologies (ICT) to be more intelligent and efficient in the use of resources, resulting in the cost and energy savings, improved service delivery and quality of life, and reduced environmental footprint—all supporting innovation and the low-carbon economy”. The origins of this particular group of definitions can be attributed to initial conceptualizations of SCs, where, following Alawadhi et al., initial definitions stressed the significance of new ICT with regard to modern infrastructures [8] .

Gradually these technically oriented definitions have proven themselves to be conceptually somewhat too narrow, urging the academic community to start emphasizing other aspects, for instance a more governance-oriented approach which emphasizes the role of social capital and relations in urban development [5] . 

In this respect, the third group takes into account the aspects of the first and the second group, but underscores the importance of (open) data collection of capturing live real world data through devices like sensors, meters, appliances, personal devices, etc., although not forgetting about complex city’s social dynamics, especially in terms of its people and their needs within the context of pursuing functioning, efficient smart sustainable cities. In this vein, this definition puts forward a socio-technical view of the smart city and underscores Beck’s argument [9] on smart cities which urges for more political awareness among researchers and developers of information and communication technologies (ICTs).

Following Nam and Pardo [10] (p. 288), this perspective puts forward a more efficient way to seek solutions for various challenges and problems encountered in modern cities due to its “comprehensive understanding of the complexities and interconnections among social and technical factors of services and physical environments in a city”, underlining the importance of exploring both, the role of technology in changing the city, and the role of the needs of communities living in urban environments. 

This group of definitions of the proposed taxonomy on the SCs stresses the need to translate theoretical, almost normative concepts of the SCs to “praxis” or empirical dimension. This somewhat neglected aspect in the SCs debate is perhaps best encapsulated by Visvizi and Lytras [8]. They are defining this lack in the discourse on SCs as a connection between the normative and the empirically feasible. To overcome this neglected aspect, Visvizi and Lytras [8] have designed a nested clusters model by which they are emphasizing the intimate relation between individuals’ well-being, their active civic engagement and SCs sustainability, sustainability being a function of the SCs’s structural clusters and their integration along processes of policymaking and strategy considerations. We agree with the pragmatic and demand-driven SCs research of Visvizi and Lytras [8], which could very well be applied within the SVs context as well. However, in our research, that requires the use of the IoT (technology) in SCs and SVs in terms of similarities and differences, we have focused more on the design of the digital innovation ecosystem; its ability to encompass different, yet interconnected fields, which are creating services that are of benefit for the citizens, are putting to the foreground the importance of the concept of the public value, which we regard—similar to Chamoso et al. [3]—as the ultimate goal of a smarter city. 

### 3.3. Public Value as the Proposed Conceptual Architecture of Digital Innovation Ecosystem of Smart Cities 

O’ Flynn [12] (p. 358) argues that “public value [is] as a multidimensional construct—a reflection of collectively expressed, politically mediated preferences consumed by the citizenry created not just through ‘outcomes’ but also through processes which may generate trust or fairness”. Accordingly, the concept of public value is complex and to refer to Chamoso et al. [3] (p. 3) includes several dimensions: (i) creation of economic and social values, both difficult to unite and sometimes enter into conflict with each other; (ii) generating value for different stakeholders, with a possibility of having different expectations not always compatible with each other (iii) creation of value in regard to the different dimensions of life in the city, implicating understanding what the real needs and priorities are. Correspondingly, to ensure public value in a smart city, the complex dynamic of a smart city’s structural framework needs to be taken into account, i.e., a large set of variables, where definition of citizens’ priorities, needs, and expectations of daily city life with their accompanying technical domains of use needs to be defined. Although Chamoso et al. are including the dimension of environment within the third set of definitions on public value in connection to possibilities of application of SC technology in a wide range of aspects of daily city life, we would like to expand Chamoso, González-Briones, Rodríguez, and Corchado’s definition of public value by adding to the presented definition the dimension of environmental value. To refer to Bulkeley et al. [13], the relation between environment and smart city initiatives has become even more urgent with recent studies of climate change, urban transitions to low carbon output [14], and increased discussions about eco or green cities as smart [15]. Madlener and Sunaare [16] stated that research on energy use and urbanization demonstrates that two thirds of the world’s total energy consumption and 70% of C02 emissions are urban, putting environmental concerns to the forefront of the smart cities initiatives. Viitanen et al. [17] are stressing that the smart city can be understood as an urban strategy that seeks to provide advanced technological solutions to many of today’s urban challenges. However, following Hamann et al. [18], inequality is one of the key social challenges of our time, and the biosphere is far from being independent, generating a need to develop understanding of the smart city, which grounds SC technology within a wider social, political, economic, cultural, environmental, and organizational context. 

In this manner, our understanding of SCs goes beyond assumptions that ICT can automatically make cities more economically prosperous and equal, more efficiently governed, and less environmentally wasteful. Contrary to such views, just like Chamoso et al. [3] we are opting for the version of the smart city based on the premise of public value, which understands cities as complex and nonlinear systems, meaning that their attendant problems should be tackled by open data and publicly transparent participatory technologies and practices, stressing the needs of ordinary people and communities that make up cities. In this manner, we place technology within and outside of the SCs digital innovation ecosystem, by outside being understood as the public value and technology as being in the function of providing public value in serving the needs, perceptions and expectations of the citizens; technology is therefore socialized in the name of public value. However, identifying a specific role of ICT in the smart city as such a complex system requires an in-depth understanding of the specific city context, as well as a firm grasp of the vast array of roles for ICT in delivering value to the city [19]. Yet to refer to Visvizi and Lytras [8] and Cosgrave et al. [19], these two core dimensions are unfortunately rarely found in the same place, and necessitate effective cross-sector engagement, dialogue and action. This means that an approach needs to be proposed that encompasses transdisciplinary thinking since diverse capabilities and knowledge is needed to allow addressing the many challenges for urban as well as rural settlements, arising from the rapid social and environmental changes. In this text, we are thus suggesting undertaking the approach of co-production as it refers to participatory or collaborative governance based on the already indicated transdisciplinarity.

### 3.4. Co-Production as a Participation-Based Practice in Digital Innovation Ecosystem of Smart Cities 

Ruiz-Mallén [7] is defining the concept and practice of co-production as an example of participatory or collaborative governance for providing legitimacy for the measures implemented by policymakers to tackle climate change. However, we argue that this approach can be applied within the context of SCs initiatives, based on the premise of public value, and even within other areas of technological innovation, science, policy, and society as well—SVs for example. 

Co-production is about participatory process that supports stakeholders’ engagement in planning and management decision-making and can be referred to government, researchers, and community actors, but also to market and third sector actors [7], since it “recognizes the relevance of having a diversity of societal actors involved in the planning and decision-making [7] (p. 2). Via Arnstein, Ruiz-Mallén [7] (p. 3) states that “co-production is thus expected to entail a certain degree of citizens’ power through establishing a “partnership” between laypeople and traditional decision-makers (i.e., government), enabling non-expert citizens to engage in decision-making through contributing their knowledge and capacities”. 

Precisely this combination of different stakeholders and facilitation of participatory practice can assure transparent technological development and results in building digital innovation ecosystems, based on public value, from smart cities to smart villages.

## 4. Digital Innovation Ecosystem of Smart Villages 

EUROSTAT defines rural areas as areas which are thinly populated, whereas urban areas are divided into two subgroups: towns and suburbs/small urban areas with intermediate density areas and cities/large urban areas with densely populated areas [20]. However, we would like to expand this definition of rural areas by adding the definition from The international association Rurality–Environment–Development (R.E.D.) [21] which emphasizes the notion of rural territory, attesting cooperation between rural, urban, and peri-urban territories, and within metropolitan areas, between urban and rural poles, stressing the importance of going beyond narrow geographical or statistical assessments, being critical to limiting itself to concept of agricultural and natural spaces and above all, underlining the element of diversity. This R.E.D. definition is important, as the indicated systems regarding the application of IoT technology have slightly different requirements to follow Kaur [2]. If application of IoT technology in SCs can be characterized by densification of IoT to day-to-day life, following cities’ structural characteristics of being densely settled places, the use of IoT technology in the SV’s digital innovation ecosystems can be associated with dispersion and deficiency. 

It is important to stress that in this text we will in particular focus on rural areas in Europe. A more detailed geographical specification is important, as to refer to Zavratnik et al. [4], there are considerable differences when dealing with rural areas in Europe and elsewhere, arguing that “the main difference is that, in Europe, the basic infrastructure is already established, whereas, in some other “un-developed” regions, the infrastructure is yet to be established” [4] (p. 11).

Based on EUROSTAT [20] in 2018 about 29.1% of the EU-28 population lived in rural areas across Europe, urging politics and research not to take into account only challenges of SCs, but also focusing on rural regions to develop accurate understanding of the needs of these communities, as it is not sufficient to apply modern ICT and use existing concepts in SCs to SVs and “make them smart”. To paraphrase Hosseini et al. [6], efforts must be extended in such a way that communities can attract and advance their own innovative potential, preventing the risk of poverty or social exclusion.

If “the most representative features of the SC are shared ICT structures, time optimization, open government, energy efficient technologies, reduced emissions, and orientation towards green environment” [4] (p. 3), the transition to smart infrastructure in rural areas and communities as of more sparsely inhabited areas is, according to Zavratnik et al. [4], even more complex and necessary. 

However, the point of sparsity of rural areas and settlements represents the central conceptual point of this research, as with this text we want to evaluate the role of the IoT (technology) in a digital innovation ecosystem of SV, adding that this ecosystem is usually a system of dispersion and deficiency, whereas the digital innovation ecosystem of the SC is about densification of the IoT (technology) to day-to-day life, following cities’ urban characteristics of being densely populated areas. 

Following Provenzano et al. [5], recent studies show a great importance of smart strategies and innovation in rural settlements, especially within the context of the widening gap between thinly and densely populated areas. On the other hand, as Hosseini et al. [6] are arguing via Porter et al. [22], these areas have enormous economic potential. We would like to add that this potential is not just economic, as it is social as well. To put in the context of the European Union, in the Cork 2.0 Declaration “A Better Life for Rural Areas” from 2016, “potential of rural areas and resources to deliver on a wide range of economic, social, and environmental challenges and opportunities benefitting all European citizen” [23] is given. Yet, delivering smart infrastructure into these areas needs to take into account specific characteristics and challenges when compared to urban environments. 

Only recently the concept of SVs has been introduced within the European Union, specifically, Smart Village Initiative was launched by the European Parliament in 2017. In documents prepared by European Network for Rural Development, we can read that the concept of Smart Villages is a relatively new policy concept in Europe and “has the potential to both add to and build upon this existing track record and create further synergies between the funds” [24]. However, the concept of SVs needs to be also conceptualized within a global context, as there are various initiatives promoting the concept of smart villages [4].

If, according to Zavratnik et al. [4], global initiatives are more focused on the areas with the lack of basic infrastructure like electricity, water supply, internet access, etc., the European initiatives are working with communities where basic infrastructure is already existing, therefore “addressing different challenges of smart and sustainable development through products and services with social, economic, and environmental benefits” [4] (p. 5). The differences within the application of the concept of SVs attest their conceptual variability and applicability, clearly demonstrating that there is no such thing as rural areas’ uniformity [4]. Based on this argument, Zavratnik et al. [4] are suggesting taking a place-based approach when working on digital transformation of the SVs. 

### Introduction of Place-Based and Context-based Approach and Possibilities for Co-Production as a Participation-Based Practice in Digital Innovation Ecosystem of Smart Villages 

Via this text we are claiming that place-based and context-based approach seems in particular useful in applying the SC concept to rural areas, as it puts forward “bottom-up integrated approaches”, by which “communities put themselves behind the steering wheel and not impose developmental paradigms that would not be compatible with community’s desires and cultural environments” [4] (p. 3). To continue with Towers [25] (p. 220), the bottom-up approach is advocated by community action and is associated with the development of local democracy. 

A place-based approach is designed to target unique conditions of specific environments, enhancing collaborative decision-making processes, sharing initiative, fostering local talents, resources, and stakeholder interests [26]. 

Above all, the place-based and context-based methodological approach has the ability to address complex needs of communities, and therefore seems particularly relevant when contextualizing the differences and similarities of the IoT empowered ecosystems of SVs and SCs, as it allows to target structural specifics of both digital innovation ecosystems very precisely. 

However, these approaches can only be successful if they are supported by policy makers. In this manner, the above indicated approach of co-production presents itself as a viable option, as it is based on “partnership” between laypeople, market, and third sector actors and traditional policy makers (i.e., government), enabling non-expert citizens to engage in decision-making through contributing their knowledge and capacities. Contrary to some of the challenges co-production faces in SCs particularly in terms of being a time-consuming process [7] (p. 9), this approach seems a lot more feasible in the digital innovation ecosystem of SVs, especially if synergies with the place-based approach are established. 

## 5. From Internet of Things (IoT) Empowered Digital Innovation Ecosystem of Smart Cities to IoT Empowered Digital Innovation Ecosystem of Smart Villages

A quick overview of the research done on SCs and SVs shows that both are not just about solving complex problems through technology, but, to paraphrase Sutriadi [27], are also about enhancing local knowledge, making technology their constitutive part. Further, we are emphasizing the importance of open data and bottom-up integrated approaches within processes of digital transformation, underscoring the importance of ensuring the dimension of public value. 

In this text we will demonstrate that in the IoT technology domain the use of IoT architecture components are application-specific alongside with acceptance that contributing factor for connectivity is based on a difference in topographical features between the two ecosystems. Technical characteristics in the IoT application domain of each ecosystem will be exhibited by referencing use-cases which include distinct service fields as these two classifications do not overlap, rather they complement each other.

### 5.1. Overview of the IoT Technology Application in Digital Innovation Ecosystem of Smart Cities

Alavi et al. [1] are claiming that the IoT technology is the building block for next generation smart cities due to its potential in exploiting sustainable information and communication technologies. As argued by Kaur [2], the concept of SCs has originated from the Internet of Things technology, due to its ability to provide efficiency in implementation in everyday life. To refer to Mazhar Rathore et al. [28], there are different domains in which IoT plays an important role in the cities, consequently improving the quality of a city life—from health care, automation, transportation, emergency response and city’s resilience to manmade and natural disasters, and even urban planning. Nonetheless, to address similarities as well as differences of the IoT application in digital innovation ecosystems of SCs and SVs, we will provide a closer look into the areas of application, starting with SCs.

The expression Internet of Things (IoT) was first described in 1999 as the emerging Internet-based technology [29]. Driven by the technology advancement rather than user needs or market demands, the IoT (technology), through pervasive use of ICT, has been interconnecting physical and virtual things [1] at rapid growth and has become a central feature of smart cities.

Today, numerous definitions of the IoT (technology) exist. According to The European Technology Platform on Smart Systems Integration [30], the IoT (technology) is defined as the “the network formed by things/objects having identities, virtual personalities operating in smart spaces using intelligent interfaces to connect and communicate with the users, social and environmental contexts. Things having identities and virtual personalities operating in smart spaces using intelligent interfaces to connect and communicate within social, environmental and user contexts”. 

To refer to Kima et al. [31], IoT applications are encouraging SC initiatives worldwide, indicating that they in particular are standing out as the most prominent IoT application. Regarding the use of the IoT (technology) in SCs, referring to Perera et al., [32] the relation between IoT technology and smart cities is the relation of ensuring provision of better services for contemporary cities. According to Alavi, et al. [1], SCs consist of six major components, i.e., of smart governance, smart economy, smart citizens, smart mobility, smart environment, and smart living. Within these components, which can also act as indicators for SCs, areas of IoT application domains come to the foreground. 

In our text we will propose the following IoT application domains, which will also serve as a base for research on smart villages: 1. Natural Resources and Energy, 2. Transport and Mobility, 3. Smart Building, 4. Daily Life, 5. Government, and 6. Economy and Society.

### 5.2. Overview of the IoT Technology Application in Digital Innovation Ecosystem of Smart Villages

On the other hand, to follow Kaur [2], the idea of the IoT (technology), combining benefits of multiple technologies, to enhance the quality of life in a city via intelligent devices, can be extended to the villages as well, demonstrating that the IoT (technology) is not limited to either context or does not exclude the understanding that there exists a difference in priorities of planning these solutions on distinct levels and contexts. 

Nevertheless, the main differences are not so much coming from the technology itself, as they are coming from socio-economic, cultural, and policy specifics of particular types of digital innovation ecosystems, demonstrating regions’ specifics and existence of significant gaps between the design and the implementation of digital transformation. As already indicated above, in Europe’s rural areas the basic infrastructure is already established, whereas, in some other “un-developed” regions, the infrastructure is yet to be established [4]. In terms of designing digital innovation ecosystem of the SV, a carefully planned strategy, resonating IoT systems architecture of identification, sensing, communication, computation, services and semantics (illustrated in Figure 1) on one hand, and working with communities via participatory, place-based, and context-based approaches on the other hand need to be taken into account. 

### 5.3. IoT Technologies Domains

Several studies have made an attachment to IoT taxonomy of application and technology domains [1]. The IoT technologies domains are focused on technical layers of distinct measurement equipment, data-flow, feedback loops, logical processing, and outputs inside such systems, in lieu of the IoT application domains convene the “outer” results and the impact on distinct society and infrastructure domains. Distinct types of technologies (sensors, APIs, databases, actuators, platforms, portals…) used within IoT solutions are not exclusively limited to differences in IoT in urban and rural settings, rather than specific functional design of the solutions. With regard to this, we will only mention different types in the passage, to familiarize a portion of the readers with building blocks used in the IoT solutions architecture. Staying cognizant, despite numerous ongoing efforts, the digital divide continues to cause a huge gap in prevalence and availability of broadband connectivity between rural and urban areas. This has been a historical hurdle and continues to be one of the largest technical obstacles in implementing smart agriculture and precision farming in practice. Additionally, distinct differences in connectivity requirements exist between urban and rural environments, e.g., IoT in rural typically requires solutions that enable large area coverage as in the city, where scenarios allow for efficient close-range solutions [33].

The 5th generation (5G) of mobile technologies seems promising in this respect. Unlike 3G and 4G, the deployment of which was clearly city-centric, introduction of 5G can take a more uniform and distributed approach. In technical terms, this can be achieved based on several enabling concepts introduced with 5G, including the support of Massive Machine-type Communication (mMTC) designed to support connectivity of large deployments of devices and sensors with ultra-low latency guarantees and exploitation of Multi-Access Edge Computing (MEC), and radio access network densification, software defined networking, massive Multiple Input Multiple Output (massive MIMO) techniques and the use of higher frequency bands that could possibly be dedicated for private industry-specific and location-dependent 5G deployments. At the same time, technical capabilities and adaptable private deployment architectures pave the way for 5G to approach its rollout in a new way and develop innovative business models that were not possible in previous generations, and with that also creation of an impact on deployment strategies in particular for sectors, such as Smart Agriculture. 

Classifications in this aspect of IoT technologies encompasses comparable items [1]. Al-Fuqaha et al. [9] comprehensively present these components of IoT systems architecture: Identification, Sensing, Communication, Computation, Services, and Semantics (illustrated in Figure 1).

Identification—Different identification methods provide a clear and unique identity for each important object within the IoT architecture. Identity management in the IoT environment needs to be able to distinguish devices, sensors, monitors and control their access to sensitive and non-sensitive data. We must distinguish between an object’s identity (e.g., EPC naming, uCode, MAC address of an interface controller) and its network address. Since we are discussing the Internet of Things, the most well-known addressing mechanism in function of the network is, for example, IPv4/6 [9].Sensing—Sensing is gathering any information (physical or digital) from connected objects and forwarding it to data warehouse, database, or cloud [9]. Sensing is enabled by a vast extent of different sensors, for example: smart utilities meters, dust particles environment readings, CO2 sensors, sunlight, exposure and radiation sensors, hydrostatic pressure level sensors, automotive and traffic sensors, navigation and GPS services, security cameras, motion sensors, card readers and door access control, distinct agricultural, crop and livestock sensing, sensors related to health services, household device sensors, industry production process related sensors, various wearable sensors, actuators, and many more [34].Communication—The implementation of IoT-based smart cities infrastructure depends significantly on efficient short- and wide-range communication protocols to transport data between sensors, devices, aggregators, data storage and processing nodes. Examples of IoT supporting communication technologies entail RFID, NFC, UWB, Bluetooth, BLE, IEEE 802.15.4, Z-wave, WiFi, LTE (Long-Term Evolution based on GSM/UMTS mobile network technologies) with support for Narrow Band IoT capabilities (NB-IoT), LoRaWAN [9], [34] as well as the emerging 5G (Fifth Generation) mobile technologies supporting massive machine-type communications (mMTC) [35].Computation—Processing hardware and software represent the “smart” in IoT. Although clever IoT architecture can address certain efficiency and coherence in logical processes, the dedicated computation devices are the “brain” of IoT. We can identify several types of equipment that support IoT: hardware nodes that run IoT applications (besides computers and smartphones also Arduino, Raspberry Pi, UDOO, FriendlyARM, Intel Galileo, Gadgeteer, BeagleBone, Cubieboard, Z1, WiSense, Mulle, T-Mote Sky...), software (e.g., smart city open source platforms as FIWARE, OCEAN, OM2M, Contiki, ODL IoTDM) [34], cloud (e.g., Hadoop), and other distributed data storage or fog, edge computing concepts [9].Services—Basically, IoT services can be classified as one of 4 types: identity-related, information aggregation, collaboration-aware, and ubiquitous services. Most rudimental, identity-related services enable the definition of real-world objects to the virtual representation inside IoT applications. Further information-aggregation collects and summarizes measurement equipment signals in IoT application. Collaborative-aware services complement obtained data through reactive decision mechanisms. Ubiquitous services are future promise of omnipresent, available anytime anywhere applications [34].Semantics—The ability for knowledge extraction with using resources, modeling data, analyzing, recognizing patterns, and presenting the information to make sense and provide with exact service [9]. The heterogeneity of IoT elements is a challenge in terms of interoperability although there lies an opportunity in this ontology as, possibly dynamic IoT architecture, as a set of node properties, relations and interactions between them, define an interesting subject area, moreover newly discovered ontologies provide a basis for better problem solving [36].

Major features of the IoT architecture, as a whole, are security (integration standards and integrity of data), privacy (encryption and cryptography), trust (decentralization avoiding single point of failure), risk management (threat modelling and efficient risk decision mechanisms), interoperability (integration in vendor locked-in services, open reference models), low-power and low-cost communication (battery life and advancements in micro-electronics), big data (analytics performance), connectivity (mobility, wide range of IoT devices and mechanism in case there is low-quality or absence of communication networks) [34].

### 5.4. Standards and Indicators in Digital Innovation Ecosystems of Smart Cities and Smart Villages

In light of associating both ecosystems with one another, we have to expound the standardization endeavors with major organizations, such as International Organization for Standardization (ISO), International Telecommunication Union (ITU), European Telecommunications Standards Institute (ETSI), United Nations (UN) and Institute of Electrical and Electronics Engineers (IEEE), to achieve common perspective in regard to relevant standards, data and common performance indicators. However, it is not our objective to solidify SCs and SVs baseline standards specification in this text, insomuch we want to query where distinct standards fit and if they can put substance on SCs and SVs, principally in terms of the IoT (technology).

Standards and indicators set a complex phenomenon in a form that is easier to quantify, understand and communicate. For the usable overview we adopt the British Standards Institute (BSI) framework that describes smart city standards and activities on strategic, process and technical specifications levels [37,38] and try to combine them with comparative analysis of standardized indicators for smart sustainable cites [39] as it includes the notion of smartness and sustainability, which need to be understood as major components of the SV debate.
Strategic standards and activities aim to support overall smart city strategies by identifying priorities, developing plan and enable effective monitoring and evaluating progress. These include:
○ISO 37120:2018 Sustainable cities and communities—Indicators for city services and quality of life [40] contains 104 indicators with test methods to measure performance management of city services and quality of life over time; transfer of results, allowing comparison across a wide range of performance measures and support of policy development and priority setting.○ISO/DIS 37122:2018 Sustainable development in communities—Indicators for smart cities [41] with 85 indicators complements ISO 37120:2018 and establishes indicators with definitions and methodologies.○ISO 37105:2019 Sustainable cities and communities—Descriptive framework for cities and communities [42] to support city and community stakeholders to define a common language to describe cities and communities and form an ontology for planning and implementing city, operating solutions that might include digital machine-readable information.○ISO 37100:2016 Sustainable cities and communities—Vocabulary [43] defines terms relating to sustainable development in communities, smart community infrastructure and related subjects.○ITU-T Y.4902/L.1602 key performance indicators related to the sustainability impacts of information and communication technology in smart sustainable cities [44] with 30 indicators describing topics such as environmental sustainability, productivity, quality of life, equity and social inclusion, physical infrastructure.○ITU-T Y.4903/L.1603 key performance indicators for smart sustainable cities to assess the achievement of sustainable development goals [45] entail 52 indicators and construe economy, environment, society, and culture topics.○Sustainable Development Goals 11+ monitoring framework [46] with 18 indicators aimed at achieving UN Sustainable Development Framework targets.Process standards and activities provide guidelines for managing smart city projects:
○ISO 37101:2016 Sustainable development in communities—Management system for sustainable development—Requirements with guidance for use [47].○ITU-T Y.4901/L.1601 key performance indicators related to the use of information and communication technology in smart sustainable cities [48] has 48 indicators and describes the categories of environmental sustainability, productivity, quality of life, equity and social inclusion, physical infrastructure related to information and communication technologies.○2413-2019 IEEE Standard for an Architectural Framework for the Internet of Things (IoT) [49] conforms to previously defined standards to congregate IoT system’s stakeholders across multiple domains (transport, healthcare, Smart Grid, etc.).Technical specifications standards describe the solution implementation level:
○P2510 IEEE Standard for Establishing Quality of Data Sensor Parameters in the Internet of Things Environment [50] defines quality measures, controls, parameters and definitions for sensor data related to Internet of Things (IoT) implementations.○P1451-99 IEEE Standard for Harmonization of Internet of Things (IoT) Devices and Systems [51] defines methods for data sharing, interoperability, and security of a network, where sensors and other devices interoperate.○ETSI Technical specification 103 463 key performance indicators for sustainable digital multiservice cities [52] entails 76 indicators and describes different sustainability related themes such as people, planet, prosperity, governance.

These are the standards, indicator sets, and activities directed at the SC. We acknowledge that the relevant standards are spread across different domains, scopes, and approaches; in addition to relevant SC items, the notions of smartness and sustainability are associated with all three levels (strategic, process, and technical) and could be also used in the context of the digital innovation ecosystem of the SV. 

### 5.5. A Proposal of a List of IoT Application Domains: Distinct Service Fields with Use-Cases

Researching literature, technical reports, and popular articles in IoT solutions, architecture, and approaches in the context of smart cities provides usable structure in means of solutions domain. Further research on IoT technology used in rural areas, the prevalent spatial focus of smart villages concept, contributes to the comparison among different domains which some are dominant in the smart city and other in the smart village context. To profess the importance and provide more traction on particular problem/solution domains we naively classify these areas and connected use cases in distinct context (SCs/SVs). We adopted the classification promising most general overview of the IoT application [3]. Table 1 below serves to exemplify that the use cases described in literature, as a part of a particular domain, can be found in both SCs and SVs and IoT contexts, thus exposing the similarities rather than disparity between them.

An immense quantity of academic resources available on IoT (technology) on SCs exists. The majority of broad spectrum of solutions in SCs is fortunately served in distinguished state-of-the-art review article by Pablo Chamoso et al. [3] and other sources [1]. As there is less apparent focus on SVs in academia we use disparate resources to complement the list (as listed in Table 1 with accompanying references), while recognizing different levels of depiction which however does not enable us with direct comparison between related use cases in terms of implementation.

## 6. Discussion

This research was grounded on two central research questions, epistemically based on transdisciplinary methodology which proves itself to be the most effective in meeting the many challenges, coming from working with introduction of technology to communities. 

To adequately cover the complex sociopolitical, cultural, and economic dynamics of SCs and SVs, the research introduced the concept of the digital innovation ecosystem, stressing that both have their own structural socio-economic features as well as geographical distinctions. 

Based on the work conducted, we can assess that the IoT (technology) is not limited to either context or does not exclude the understanding that there exists a difference in priorities of planning these solutions on distinct levels and contexts, thus providing this research with two key interconnected conclusions: Exemplified in Table 1, general similarities rather than disparities exist between both digital innovation ecosystems (technology, domains, and standards).As the study has shown that the same IoT (technology) can be used in both digital innovation ecosystems, the addressed question of potential differences in terms of the use of the IoT (technology) in both digital innovation ecosystems has to do more with sociopolitical and cultural dimensions than with technology itself, demonstrating that technology is far from being a neutral phenomenon.

Following the second conclusion, unique characteristics of both digital innovation ecosystems come to the foreground: if a digital innovation ecosystem of SCs can be characterized by its higher level of density in terms of the key elements of the digital innovation ecosystems (tech end-users, value chain actors, business intelligence, competence centers, big industry, investors, tech providers, tech adopters, public authorities, universities, SMEs, incubators, research organizations, start-ups and citizens), a digital innovation ecosystems of the SVs obviously has a structure of lower density of the above mentioned key elements. Moreover, these elements are different and demand different strategic approaches. 

Because this work has clearly demonstrated that technology cannot be examined in separation from the sociopolitical, economic, and cultural context, it is rather important how collaboration with the communities is established and led. This issue encompasses multiple levels of the innovation digital ecosystems: policymaking, interaction between citizens and technology, innovation as (public) value, economy as sustainable economic growth and in the end, basic social awareness in terms of the sustainability of the digital innovation ecosystems themselves. 

As motivation for this research came from our direct involvement in the fieldwork research activities carried out within domestic and international projects, focusing on both digital innovation ecosystems, the research findings will be further applied in the above mentioned projects, exhibiting future prospects and trends of this research. 

In line with creating best possible approaches for working with the communities, we will directly test the proposed approach of co-production, although, following conducted transdisciplinary analysis, modified for each digital innovation ecosystem. 

Furthermore, to overcome the so-called normative bias of the ICT in SCs and SVs research discourses, currently we are working on the Meet the Local Producer platform, where up to 5 different agricultural holdings will be engaged for the purpose of carrying out the project activities, most of which are young farm holders. The reason for this is that the Slovenian Rural Development Program clearly states that the age structure of agricultural holders has a significant impact on agricultural labor productivity, as young agricultural holders are more motivated for life-long learning, implementing technology transfer and innovation as well as in adapting the farming practices to environmental and climate change challenges.

By introducing the use of new IoT technologies in agriculture but especially in viticulture, where technology assures lower production costs compared to conventional way of farming in the range between 20-30%, economic efficiency will be ensured, making it an opportunity especially in terms of optimizing the costs of irrigation, use of fertilizers and vineyards chemical protection as well as the costs of labor. 

In addition, social implications generated by this particular activity underscores pragmatic, demand driven approach, which follows the needs of communities, especially from rural and peripheral areas. It highlights the premise of public and environmental value, as on one hand it directly works for citizens’ good, and on the other hand, it enables the IoT (technology) enhanced services.

## 7. Conclusions

To effectively emphasize characteristics of SCs (urban settlement) and SVs (rural settlement), this paper has proposed the concept of a digital innovation ecosystem via which complexity of digital transformation with all its accompanying phenomena, including the application of the IoT (technology), has been underlined in both digital innovation ecosystems.

Further analysis based on the concept of the digital innovation ecosystem has revealed different aspects and focal points of presented systems: each ecosystem has their own structural socio-economic features as well as geographical distinctions, which need to be taken into account when designing IoT solutions. To generalize, SCs are characterized by densification of the technology (IoT) while SVs epitomize a system of dispersion and technology deficiency. 

The evocation of the concept of an digital innovation ecosystem has demonstrated that broad involvement of policy makers, researchers and community actors, market and third sector actors within both ecosystems present a key component in the strategy process of successful harnessing of the IoT (technology) in terms of the public value in both ecosystems. In this regard, this research has proposed building a transdisciplinary partnership based on a participatory process of co-production which facilitates “bottom-up” approaches. 

In the IoT technology domain, the use of IoT architecture components are application-specific alongside the acceptance that the contributing factor for connectivity is based on a difference in topographical features between the two ecosystems. Structural characteristics in the IoT application domain of each ecosystem are exhibited by referencing use-cases which include distinct service field (e.g., Smart Building, Smart Agriculture, E-health, Smart Grids, Smart Mobility, etc.). 

The OECD has recently estimated that there will be as many as 25 billion devices connected to the Internet by 2020 [74]. IoT supported SCs and SVs as socio-technical ecosystems alongside proper planning and implementation approaches, acknowledging social and political concerns about transparency and accountability of the IoT algorithmic processes [75], can provide a balance between economic growth, social well-being and care for the environment without compromising the capacity of future generations. As IoT (technology) is still an emergent field, transparency with respect to open data [75] and creation of public value is necessary, also in regard to stipulation of human centered digital transformation which is about facilitation of the type of digital transformation, that pursues complementary, systematic, structured and holistic approach, being very much aware of its ramifications in shaping citizenship within data ecosystems [76], government, commerce, and personal privacy. 

## Figures and Tables

**Figure 1 sensors-20-03897-f001:**
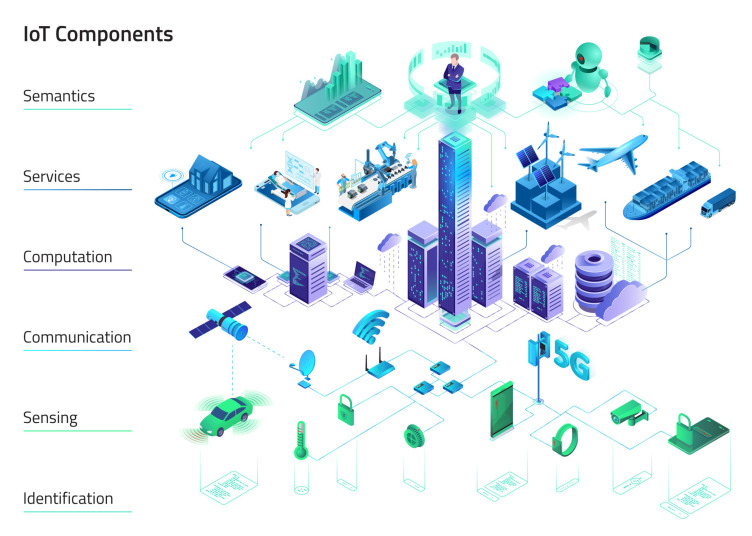
Internet of Things (IoT) Architecture Components.

**Table 1 sensors-20-03897-t001:** A Proposal of a List of IoT Application Domains: Distinct Service Fields with Use-Cases.

Domain [3]	Smart Cities Use Cases	References	Smart Villages Use Cases	References
Natural Resources and Energy	Smart Grids, Smart Public Lightning, Renewable Energy, Waste Management, Water Management, Food and Agriculture	[1,3,53,54]	Smart Weather and Irrigation, Smart and Precision Farming, Energy Access, Food Security, Micro Smart Grids	[55,56,57,58,59,60,61,62,63,64,65,66]
Transport and Mobility	City Logistics, Smart Mobility Information and Options, Parking Solutions	[1,3,54,65,66]	Smart Mobility	[65,67]
Smart Building	Facilities Management, Construction Services, Housing Quality	[1,3,53,68]	Smart Buildings	[56]
Daily Life	Entertainment, Hospitality, Pollution Control, Public Security, E-health, Welfare and Social Inclusion, Management of Public Spaces	[1,3,9,57,69,70],	Smart Healthcare, Smart Surveillance System	[56]
Government	E-governance, E-democracy, Transparency	[3,58,59]	Smart Elections	[71]
Economy and Society	Innovation and Entrepreneurship, Cultural Heritage Management, Digital Education, Human Capital Management	[3,72,73]	Smart Education	[56,71]

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
