# Peer review of "The Use of IoT Technology in Smart Cities and Smart Villages: Similarities, Differences, and Future Prospects"

_sensors, 2020, doi:10.3390/s20143897_

Round 1

Reviewer 1 Report

This manuscript is a good survey of IoT systems in smart cities and villages. It is well written, and easily readable for a large audience.

My major remark is that no standardization activities are mentioned, while there are a certain number carried out by both ISO and IEEE. Among others: ISO 37100, ISO 37105, ISO 37120, IEEE P1451, IEEE P2413, IEEE P2510.

The manuscript should be revised and these important activities included, even because it is not possible to compare different scenarios if data are not standardized.

Author Response

Dear reviewer, 

Thank you for reviewing the manuscript. We greatly appreciate your comments and recommendations. We have tried our best to revisit the manuscript accordingly.

Our response to your comment can be seen bellow.

  1. Comment:

“Comments and Suggestions for Authors

This manuscript is a good survey of IoT systems in smart cities and villages. It is well written, and easily readable for a large audience.

My major remark is that no standardization activities are mentioned, while there are a certain number carried out by both ISO and IEEE. Among others: ISO 37100, ISO 37105, ISO 37120, IEEE P1451, IEEE P2413, IEEE P2510.

The manuscript should be revised and these important activities included, even because it is not possible to compare different scenarios if data are not standardized.”

In this manner, we have inserted the suggested standardization activities. Corrections are visible in section 5.4 Standards and Indicators in Digital Innovation Ecosystems of Smart Cities and Smart Villages, from numbers 553 – 619.

Thank you again for your review, from which we have learnt a lot.

Sincerely.

Reviewer 2 Report

The paper aims to provide an overview of the IoT applications in a rural environment. Although the paper is focused on a timely topic, there are several issues that have to be addressed:

1) An overview of related works/surveys should be included in the introduction section. In addition, the contributions of this work should be highlighted (e.g., in the form of bullets).

2) A short paragraph (at the end of the Introduction), describing the structure of the paper should be included.

3) As this is a review paper, more the references can be added. The authors may want to consider including the following relevant references:

[1] Ahmed, Nurzaman, Debashis De, and Iftekhar Hussain. "Internet of Things (IoT) for smart precision agriculture and farming in rural areas." IEEE Internet of Things Journal 5.6 (2018): 4890-4899

[2] Triantafyllou, Anna, Panagiotis Sarigiannidis, and Stamatia Bibi. "Precision Agriculture: A Remote Sensing Monitoring System Architecture." Information 10.11 (2019): 348.

[3] Kamienski, Carlos, et al. "Smart water management platform: Iot-based precision irrigation for agriculture." Sensors 19.2 (2019): 276.

[4] Plageras, Andreas P., et al. "Efficient IoT-based sensor BIG Data collection–processing and analysis in smart buildings." Future Generation Computer Systems 82 (2018): 349-357.

[5] Lytos, Anastasios, et al. "Towards smart farming: Systems, frameworks and exploitation of multiple sources." Computer Networks 172 (2020): 107147.

[6] Muhammad, Ghulam, et al. "Smart health solution integrating IoT and cloud: A case study of voice pathology monitoring." IEEE Communications Magazine 55.1 (2017): 69-73.

[7] Begum, Syeda Faiza Unnisa, and I. Begum. "Smart Health Care Solutions Using IOT." International Journal & Magazine of Engineering, Technology, Management and Research 4.3 (2017): 10-15.

[8] Al-Dweik, Arafat, et al. "IoT-based multifunctional scalable real-time enhanced road side unit for intelligent transportation systems." 2017 IEEE 30th Canadian conference on electrical and computer engineering (CCECE). IEEE, 2017.

4) In section 4, several communication technologies and domains for smart villages are mentioned. These are also used in smart cities. For example, smart health is a common scenario for both rural and urban environments. Therefore, any differences in terms of implementations and requirements should be described. The addition of tables that summarize these comparisons would be useful.

5) The main contribution of this work is somewhat weak. The paper provides a description of communication technologies and domains, but the respective discussion section is too short. Furthermore, a section on research challenges and future trends is recommended.

6) A thorough revision in terms of the use of the English language is highly suggested.

Author Response

Dear reviewer,

Thank you for reviewing the manuscript. We greatly appreciate your comments and recommendations. We have tried our best to revisit the manuscript according.

In this manner, we have prepared a list of explanations to your lucid comments on our text:

  • Comment:

“ An overview of related works/surveys should be included in the introduction section. In addition, the contributions of this work should be highlighted (e.g., in the form of bullets).”

The introduction has been revised, following your comments. In the sector of introduction we have named the central works / surveys on which our research is based on. A more detailed presentation of the works /surveys can be found in the text itself. In addition, we have put down the main contributions of the conducted research.

2) Comment:

2) A short paragraph (at the end of the Introduction), describing the structure of the paper should be included.

Thank you for this comment, for which we believe, the text will function better. The paragraph is visible in numbers: 94 -100.

3)Comment:

As this is a review paper, more the references can be added. The authors may want to consider including the following relevant references:

[1] Ahmed, Nurzaman, Debashis De, and Iftekhar Hussain. "Internet of Things (IoT) for smart precision agriculture and farming in rural areas." IEEE Internet of Things Journal 5.6 (2018): 4890-4899

[2] Triantafyllou, Anna, Panagiotis Sarigiannidis, and Stamatia Bibi. "Precision Agriculture: A Remote Sensing Monitoring System Architecture." Information 10.11 (2019): 348.

[3] Kamienski, Carlos, et al. "Smart water management platform: Iot-based precision irrigation for agriculture." Sensors 19.2 (2019): 276.

[4] Plageras, Andreas P., et al. "Efficient IoT-based sensor BIG Data collection–processing and analysis in smart buildings." Future Generation Computer Systems 82 (2018): 349-357.

[5] Lytos, Anastasios, et al. "Towards smart farming: Systems, frameworks and exploitation of multiple sources." Computer Networks 172 (2020): 107147.

[6] Muhammad, Ghulam, et al. "Smart health solution integrating IoT and cloud: A case study of voice pathology monitoring." IEEE Communications Magazine 55.1 (2017): 69-73.

[7] Begum, Syeda Faiza Unnisa, and I. Begum. "Smart Health Care Solutions Using IOT." International Journal & Magazine of Engineering, Technology, Management and Research 4.3 (2017): 10-15.

[8] Lagkas, Thomas, et al. "UAV IoT framework views and challenges: Towards protecting drones as “Things”." Sensors 18.11 (2018): 4015.

[9] Al-Dweik, Arafat, et al. "IoT-based multifunctional scalable real-time enhanced road side unit for intelligent transportation systems." 2017 IEEE 30th Canadian conference on electrical and computer engineering (CCECE). IEEE, 2017.

[10] Sarigiannidis, Panagiotis, et al. "The Big Data Era in IoT-enabled Smart Farming: Re-defining Systems, Tools, and Techniques." (2019): 107043.

Thank you for this input. These references provide with additional insights regarding the distinct domain of IoT implementation use cases within the Smart City and Smart Village context. We did not aim that this would ne a review paper, rather it is and expose of concepts that will provide us with some guidelines in future research in this field. However, we have inserted the proposed references into our text, mainly in Table 1.

4) Comment:

“In section 4, several communication technologies and domains for smart villages are mentioned. These are also used in smart cities. For example, smart health is a common scenario for both rural and urban environments. Therefore, any differences in terms of implementations and requirements should be described. The addition of tables that summarize these comparisons would be useful.”

We appreciate the comment and we agree that such comparison would be of great benefit. Unfortunately, we have dificulties comparing each related domain use cases as the resources on Smart Villages IoT implementations are very sparse, many of them not very detailed, while the Smart City technology implemetation has many detailed resources, creating predicament when comparing details of said implementations. While acknowledging this challenges, the designed table serves to exemplify the general similarites rather than disparities and gaps in IoT implementations of each context.

5) Comment:

The main contribution of this work is somewhat weak. The paper provides a description of communication technologies and domains, but the respective discussion section is too short. Furthermore, a section on research challenges and future trends is recommended.

The discussion section has been revised and research challenges with future trends / prospect have been added. Changes can be seen in numbers from 642 – 702

6) Comment:

 “A thorough revision in terms of the use of the English language is highly suggested.”

We have revisited the text again.

Thank you again for your review, from which we have learnt a lot.

Sincerely.

Reviewer 3 Report

Dear Authors, your paper sets on to explore a very important issue, yet you fail to deliver. The following points offer an insight into what could be improved in your paper:

the definition of smart village and its relation to smart cities: you just mention the 'IoT' link, yet, you may want to consider this work, in which the connection has been contextualized and conceptualized: Visvizi, A., Lytras, M. (2018) ‘Rescaling and refocusing smart cities research: from mega cities to smart villages’, Journal of Science and Technology Policy Management (JSTPM), DOI: https://doi.org/10.1108/JSTPM-02-2018-0020 ;

the point on ecosystems: what kind of ecosystems? please, improve the argument, to this end you may want to consider this paper, which offers a good set of definitions on this matter: Calzada, I. and Almirall, E. (2020), "Data ecosystems for protecting European citizens’ digital rights", Transforming Government: People, Process and Policy, Vol. 14 No. 2, pp. 133-147. https://doi.org/10.1108/TG-03-2020-0047

method and methodology: there is no section that would address this issue; please, work on it; 

structure of the paper: I don't quite see a valid research puzzle and research question; also the structure of the paper is not defined; please, address these connected issues;

the style of expression: your paper consists of a great number of direct citations collated together; this is not a good academic practice: please, rephrase and improve the style;

referencing: reference #37 something is really wrong with it; please, reconsider and revise; 

Author Response

Dear reviewer,

Thank you for reviewing the manuscript. We greatly appreciate your comments and recommendations. We have tried our best to revisit the manuscript accordingly.

In this manner, we have prepared a list of explanations to your lucid comments on our text:

  1. Comment:

“the definition of smart village and its relation to smart cities: you just mention the 'IoT' link, yet, you may want to consider this work, in which the connection has been contextualized and conceptualized: Visvizi, A., Lytras, M. (2018) ‘Rescaling and refocusing smart cities research: from mega cities to smart villages’, Journal of Science and Technology Policy Management (JSTPM), DOI: https://doi.org/10.1108/JSTPM-02-2018-0020”

The main objective of this research was to provide an examination of the role of Internet of Things (IoT) technology in SCs and SVs. To achieve this objective, this research addressed and discussed different application areas of the IoT (technology), identifying differences as well as similarities in both digital innovation ecosystems, while trying to illuminate the standardization efforts that can be applicable in both contexts. However, as we quickly found out, the best way to address this central research objective, is to tackle it from the transdisciplinary angle, therefore your suggested source was of great help. This reference is in fact used several times, to put it with numbers of the revised text: 58, 223, 227 and 276. Again, thank you for helping us with this source.

  1. Comment:

“the point on ecosystems: what kind of ecosystems? please, improve the argument, to this end you may want to consider this paper, which offers a good set of definitions on this matter: Calzada, I. and Almirall, E. (2020), "Data ecosystems for protecting European citizens’ digital rights", Transforming Government: People, Process and Policy, Vol. 14 No. 2, pp. 133-147. https://doi.org/10.1108/TG-03-2020-0047sen”

Thank you for putting this forward. Following your thoughtful instruction, we have improved the argument of the use of this concept. In this manner, we provided two central sources for our conceptualization of the argument: 1. One is based on Wang’s ecological perspective, and the other is coming from our direct involvement in an Interreg project of Carpe Digem, directly working on the concept of the digital innovation ecosystem. The revised changes can be seen in numbers of the text: 118 – 167. We hope, we have succeeded with the improvement. Regarding the suggested source; also this one is very interesting, however the concept addressed in this suggested reference is slightly different from what we have intended to do with our proposed concept. On the other hand, we have mentioned the suggested reference in number 835.

  1. Comment:

“method and methodology: there is no section that would address this issue; please, work on it;”

Thank you also for this comment. We have revised our text accordingly. Revised changes can be seen in numbers from 102-116. We have also created a specific sector, only dedicated to the issue of methodology.

  1. Comment:

“structure of the paper: I don't quite see a valid research puzzle and research question; also the structure of the paper is not defined; please, address these connected issues”

The structure of the text has been revised according to your comments. The changes are the following:

  • Revised introduction with explanation to the reader of the structure of the paper: 94-100
  • Within the sector of methodology, two central research questions have been formulated
  • The argument you had on the concept of the ecosystem was taken as a base ground for revising the manuscript. As you will see, both digital innovation ecosystems of SCs and SVs are now made as separated sections
  • To make this text better, we have also added an in-depth discussion which synthetizes the posed research questions.
  1. Comment:

the style of expression: your paper consists of a great number of direct citations collated together; this is not a good academic practice: please, rephrase and improve the style;

The used style is somehow typical for humanities and social theory – due to transdisciplinary nature of the conducted research. Where possible, the style of expression was changed according to your suggestions.

  1. Comment:

referencing: reference #37 something is really wrong with it; please, reconsider and revise; 

Thank you for noticing – we have changed it. See source 30.

Thank you for comments.

Best regards.

Round 2

Reviewer 1 Report

No further comments.

Reviewer 2 Report

The authors have appropriately addressed my concerns. I have no further comments.

Reviewer 3 Report

Thank you for addressing my duggestions